# FREQUENCY-BASED SEARCH-CONTROL IN DYNA

**Yangchen Pan & Jincheng Mei** [*]
Department of Computing Science
University of Alberta
Edmonton, AB, Canada
`{pan6,jmei2}@ualberta.ca`

**Amir-massoud Farahmand**
Vector Institute & University of Toronto
Toronto, ON, Canada
`farahmand@vectorinstitute.ai`

## ABSTRACT

Model-based reinforcement learning has been empirically demonstrated as a successful strategy to improve sample efficiency. In particular, Dyna is an elegant model-based architecture integrating learning and planning that provides huge flexibility of using a model. One of the most important components in Dyna is called search-control, which refers to the process of generating state or state-action pairs from which we query the model to acquire simulated experiences. Search-control is critical in improving learning efficiency. In this work, we propose a simple and novel search-control strategy by searching high frequency regions of the value function. Our main intuition is built on Shannon sampling theorem from signal processing, which indicates that a high frequency signal requires more samples to reconstruct. We empirically show that a high frequency function is more difficult to approximate. This suggests a search-control strategy: we should use states from high frequency regions of the value function to query the model to acquire more samples. We develop a simple strategy to locally measure the frequency of a function by gradient and hessian norms, and provide theoretical justification for this approach. We then apply our strategy to search-control in Dyna, and conduct experiments to show its property and effectiveness on benchmark domains.

## 1 INTRODUCTION

Model-based reinforcement learning (MBRL) (Lin, 1992; Sutton, 1991b; Daw, 2012; Sutton & Barto, 2018) methods have successfully been applied to many benchmark domains (Gu et al., 2016; Ha, David and Schmidhuber, Jürgen, 2018; Kaiser et al., 2020). The Dyna architecture, introduced by Sutton (1991a), is one of the classical MBRL architectures, which integrates model-free and model-based policy updates in an online RL setting (Algorithm 2 in Appendix A.3). At each time step, a Dyna agent uses the real experience to learn a model and to perform model-free policy update, and during the *planning* stage, simulated experiences are acquired from the model to further improve the policy. A closely related method in model-free learning setting is experience replay (ER) (Lin, 1992; Adam et al., 2012), which utilizes a buffer to store experiences. An agent using the ER buffer randomly samples the recorded experiences at each time step to update the policy. Though ER can be thought of as a simplified form of MBRL (van Seijen & Sutton, 2015), a model provides more flexibility in acquiring simulated experiences.

A crucial aspect of the Dyna architecture is the *search-control* mechanism. It is the mechanism for selecting states or state-action pairs to query the model in order to generate simulated experiences (cf. Section 8.2 of Sutton & Barto 2018). We call the corresponding data structure for storing those states or state-action pairs the *search-control queue*. Search-control is of vital importance in Dyna, as it can significantly affect the model-based agent's sample efficiency. A simple approach to search-control is to sample visited states or state-action pairs, i.e., use the initial state-action pairs stored in the ER buffer as the search-control queue. This approach, however, does not lead to an agent that outperforms a model-free agent that uses ER. To see this, consider a deterministic environment, and assume that we have the exact model. If we simply sample visited state-action pairs for search-control, the next-state and reward would be the same as those in the ER buffer. In practice, we have

---

[*]Equal contribution.

model errors too, which causes some performance deterioration (Talvitie, 2014; 2017). Without an elegant search-control mechanism, we are not likely to benefit from the flexibility given by a model.

Several search-control mechanisms have already been explored. Prioritized sweeping (Moore & Atkeson, 1993) is one such method that is designed to speed up the value iteration process: the simulated transitions are updated based on the absolute temporal difference error. It has been adopted to continuous domains with function approximation too (Sutton et al., 2008; Pan et al., 2018; Corneil et al., 2018). Gu et al. (2016) utilizes local linear models to generate optimal trajectories through iLQR (Li & Todorov, 2004). Pan et al. (2019) suggest a method to generate states for the search-control queue by hill climbing on the value function estimate.

This paper proposes an alternative perspective to design search-control strategy: we can sample more frequently from the state space where the value function is more difficult to estimate. We first review some basic background in MBRL (Section 2). Afterwards, we review some concepts in signal processing and conduct experiments in the supervised learning setting to show that a high frequency function is more difficult to approximate (Section 3). In order to quantify the difficulty of estimation, we borrow a crucial idea from the signal processing literature: a signal with higher frequency terms requires more samples for accurate reconstruction. We then propose a method to locally measure the frequency of a point in a function's domain and provide a theoretical justification for our method (Theorem 1 in Section 3.2). We use the hill climbing approach of Pan et al. (2019) to adapt our method to design a search-control mechanism for the Dyna architecture (Section 4). We conduct experiments on benchmark and challenging domains to illustrate the properties and utilities of our method (Section 5).

## 2 BACKGROUND

Reinforcement learning (RL) problems are typically formulated as Markov Decision Processes (MDPs) (Sutton & Barto, 2018; Szepesvári, 2010). An MDP $(\mathcal{S}, \mathcal{A}, \mathbb{P}, R, \gamma)$ is determined by state space $\mathcal{S}$, action space $\mathcal{A}$, transition function $\mathbb{P}$, reward function $R : \mathcal{S} \times \mathcal{A} \times \mathcal{S} \to \mathbb{R}$, and discount factor $\gamma \in [0, 1]$. At each step $t$, an agent observes a state $s_t \in \mathcal{S}$, and takes an action $a_t \in \mathcal{A}$. The environment receives $a_t$, and transits to the next state $s_{t+1} \sim \mathbb{P}(\cdot|s_t, a_t)$. The agent receives a reward scalar $r_{t+1} = R(s_t, a_t, s_{t+1})$. The agent maintains a policy $\pi : \mathcal{S} \times \mathcal{A} \to [0, 1]$ that determines the probability of choosing an action at a given state. For a given state-action pair $(s, a)$, the action-value function of policy $\pi$ is defined as $Q_\pi(s, a) = \mathbb{E}[G_t|S_t = s, A_t = a; A_{t+1:\infty} \sim \pi]$ where $G_t \stackrel{\text{def}}{=} \sum_{t=0}^{\infty} \gamma^t R(s_t, a_t, s_{t+1})$ is the return of $s_0, a_0, s_1, a_1, ...$ following the policy $\pi$ and transition $\mathbb{P}$. Value-based RL methods learn the action-value function (Watkins & Dayan, 1992), and act greedily w.r.t. the action-value function. Policy-based RL methods perform gradient update of parameters to learn policies with high expected rewards (Sutton et al., 1999). Both value and policy-based RL methods can be easily adopted in the Dyna framework.

**Model-based RL.** A model is a mapping that takes a state-action pair as its input and outputs some projection of the future state. A model can be local (Tassa et al., 2012; Gu et al., 2016) or global (Ha, David and Schmidhuber, Jürgen, 2018; Pan et al., 2018), deterministic (Sutton et al., 2008) or stochastic (Deisenroth & Rasmussen, 2011; Ha, David and Schmidhuber, Jürgen, 2018), feature-to-feature (Corneil et al., 2018; Ha, David and Schmidhuber, Jürgen, 2018) or observation-to-observation (Gu et al., 2016; Pan et al., 2018; Kaiser et al., 2020), one-step (Gu et al., 2016; Pan et al., 2018), or multi-step (Sorg & Singh, 2010; Oh et al., 2017), or decision-aware (Joseph et al., 2013; Farahmand et al., 2017; Silver et al., 2017). Modelling the environment dynamics through a reproducing kernel Hilbert space (RKHS) embedding has been also studied (Grunewalder et al., 2012), where the Bellman operator is approximated in an RKHS. The model we consider in this work is a one-step environment dynamics model, which takes a state-action pair as its input and returns the next-state and reward. Our proposed search-control approach, however, can be naturally used for different types of models.

The most relevant work to ours is hill climbing Dyna (Pan et al., 2019). Pan et al. (2019) proposes a search-control mechanism based on hill climbing on the value estimates (see Algorithm 3 in Appendix A.3). We briefly review the key steps of their algorithm, which is called (Hill Climbing)HC-Dyna, as it helps to understand ours. HC-Dyna maintains an ER buffer. At each step, a state is randomly sampled from the ER buffer and is used as the initial state to perform hill climbing (i.e.

gradient ascent) on the learned value function. The states along the trajectory are stored in the search-control queue.[1] During the planning stage, states are sampled from the search-control queue and are paired with their corresponding on-policy actions (i.e., actions selected by the current $Q$ network at the sampled states). Afterwards, the model is queried for each of the state-action pairs to get the next-state and reward. These simulated transitions are then mixed with samples from the ER buffer, which are observed by the agent during its interaction with the real environment, to train the value function estimator, e.g., a deep neural network.

The heuristic idea behind the search-control mechanism of HC-Dyna is that the magnitude of the value function provides useful information for guiding where to query the model. This heuristic can intuitively be understood by noticing that an RL agent tends to move towards high-value regions of the state space; by performing gradient ascent on the (estimated) value function, we provide the agent with more samples from regions where it may move towards in the future. Even if the estimated value function is incorrect and the samples are indeed from the low-value regions of the state space, these extra samples lead to the fast correction of the estimated value in those regions. Nevertheless, the magnitude of the value function is only one source of extra information from which we can design a search-control mechanism. This work suggests a different perspective: we should sample more from the regions of the state space where learning the value function is more difficult.

## 3 UNDERSTANDING THE DIFFICULTY OF FUNCTION APPROXIMATION

In a regular regression setting, we illustrate that high frequency regions of a function is difficult to approximate. We show that by assigning more training data to those regions, the learning performance considerably improves. To make this insight practically useful, we employ the sum of gradient and hessian norms of a function as a measure of the local frequency of a function. We establish a theoretical connection between our proposed criterion and the local frequency of a function. This would be the foundation of our frequency-based search-control method in Section 4.

### 3.1 WHAT TYPE OF FUNCTION IS DIFFICULT TO APPROXIMATE?

Consider the standard regression problem with the mean square loss. Given a training set $D = \{(x_i, y_i)\}_{i=1:n}$, our goal is to learn an unknown target function $f^*(x) = \mathbb{E}[Y|X = x]$ by empirical risk minimization. Formally, we aim to solve

$$f = \arg\min_{f \in \mathcal{H}} \frac{1}{n} \sum_{i=1}^{n} (f(x_i) - y_i)^2,$$

where $\mathcal{H}$ is some hypothesis space. Suppose that we can choose the distributions of samples $\{x_i\}$. How should we select them in order to improve the quality of the learned function? One intuitive heuristic is that if we know the regions in the domain of $f^*$ that are more difficult to approximate, we can assign more training data there in order to help the learning process. The important question is how to quantify the difficulty of approximating a function. We borrow an idea from the field of signal processing to suggest a method.

The Nyquist-Shannon sampling theorem in signal processing states that given a band-limited function (or signal) $f : \mathbb{R} \to \mathbb{R}$ with the highest frequency (in the Fourier domain) of $\omega_{\text{bandwidth}}$, we can perfectly reconstruct it based on regular samples (in the time domain) obtained at the sampling rate of $2\omega_{\text{bandwidth}}$ (Zayed, 1993).[2] Therefore, if the Fourier transform of a function has high frequency terms, more samples are required to reconstruct it accurately. We note that the sampling theory has been applied in the sample complexity analysis of machine learning algorithms (Smale & Zhou, 2004; 2005; Jiang, 2019). Although the problem setting in machine learning is somewhat different from this result in signal processing, it still provides a high-level intuition for us: regions with more high frequency signals require more learning data.

---

[1] According to the original paper, natural gradient is used to guarantee a certain level of coordinate invariance property, so it can handle state variables with vastly different numerical scales.

[2] Sampling rate refers to number of samples per second used to reconstruct continuous signals.

To make this high-level intuition concrete, we consider the following function:

$$f_{\sin}(x) = \begin{cases} \sin(8\pi x) & x \in [-2, 0), \\ \sin(\pi x) & x \in [0, 2]. \end{cases} \tag{1}$$

It is easy to check that the regions $[-2, 0)$ and $[0, 2]$ contain signals with frequency ratio $8 : 1$. Based on the intuition from the sampling theorem, the $[-2, 0)$ interval requires more training data than the $[0, 2]$ interval. Given the same amount of training data, and the same learning algorithm, we would expect that assigning more fraction of the training data on $[-2, 0)$ to perform better than distributing them uniformly or assigning more samples to the $[0, 2]$ interval.

**An illustrative experiment.** To empirically verify the intuition, we conduct a simple regression task, with $f_{\sin}$ as the target function. The training set $D = \{(x_i, y_i)\}_{i=1:n}$ is generated by sampling $x \in [-2, 2]$, and adding Gaussian noise $N(0, \sigma^2)$ on Eq. (1), where the standard deviation is set to be $\sigma = 0.1$. We present the $\ell_2$ regression learning curves of training datasets with different biased sampling ratios $p_b \in \{60\%, 70\%, 80\%\}$, as shown in Fig. 1 (a)-(c). We observe that biased training data sampling ratios towards high frequency region clearly speeds up learning. This is consistent with the intuitive insight and suggests that our heuristic to assign more data to high frequency regions leads to faster learning.

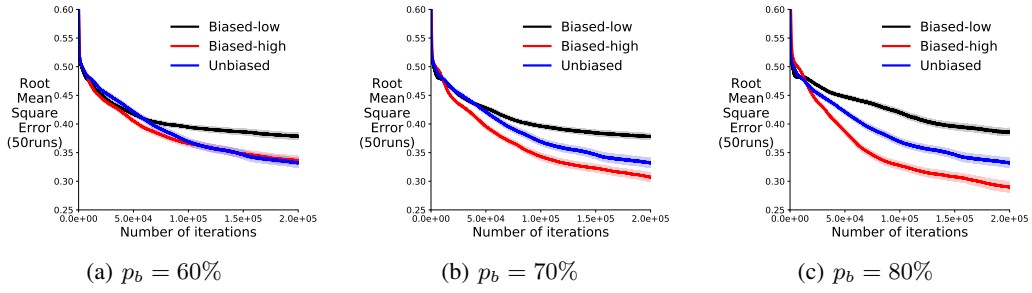

(a) $p_b = 60\%$        (b) $p_b = 70\%$        (c) $p_b = 80\%$

Figure 1: Testing error as a function of number of mini-batch updates. $p_b = 60\%$ means $60\%$ of the training data are from the high frequency region $[-2, 0)$ and is labeled as **Biased-high**. We include unbiased training dataset as a reference (**Unbiased**). The total numbers of training data are the same across all experiments. The testing set is unbiased and the results are averaged over 50 random seeds with the shade showing standard error.

## 3.2 IDENTIFYING HIGH FREQUENCY REGIONS OF A FUNCTION

Identifying the high frequency region of $f_{\sin}$ in the previous toy problem was easy, as each region contained a signal with a constant known frequency. In practice, we face two main difficulties to identify the high frequency regions of a function. The first is that we do not have access to the underlying target function, but only to data or possibly an approximate function that is estimated using data, e.g., a trained neural network. The second is that frequency is a global property rather than a local one. The value of the function at each (non-zero measure) region of the domain has impact on its global frequency representation. To make the high frequency heuristic practically useful, we need a simple criterion that (a) uses function approximation, (b) characterizes local frequency information, and (c) can be efficiently calculated. Inspired by the function $f_{\sin}$ in Eq. (1), a natural idea is to calculate the first order $f'(x) \stackrel{\text{def}}{=} \frac{df(x)}{dx}$ or second order derivative $f''(x) \stackrel{\text{def}}{=} \frac{d^2 f(x)}{dx^2}$ because they both satisfy (a) and (c). As a "sanity check" for property (b), consider the following examples.

**Example 1.** *For $f_{\sin}$ defined in Eq. (1), calculate the integrals of squared first order derivative $f'_{\sin}$ on high frequency region $[-2, 0)$ and low frequency region $[0, 2]$, respectively:*

$$\int_{-2}^{0} |f'_{\sin}(x)|^2 \, dx = 64\pi^2, \quad \int_{0}^{2} |f'_{\sin}(x)|^2 \, dx = \pi^2.$$

**Example 2.** *Let $f : [-\pi, \pi] \to \mathbb{R}$ be a band-limited real valued function defined as*

$$f(x) = \frac{a_0}{2} + \sum_{n=1}^{N} a_n \cos(nx) + b_n \sin(nx),$$

*where $a_0, a_n, b_n \in \mathbb{R}$, $n = 1, 2, \ldots, N$ are Fourier coefficients of frequency $\frac{n}{2\pi}$. Then,*

$$\int_{-\pi}^{\pi} |f'(x)|^2 \, dx = \pi \cdot \sum_{n=1}^{N} n^2 \left(a_n^2 + b_n^2\right), \quad \int_{-\pi}^{\pi} |f''(x)|^2 \, dx = \pi \cdot \sum_{n=1}^{N} n^4 \left(a_n^2 + b_n^2\right).$$

Example 1 shows that the integral of squared first order derivative ratio is $64 : 1$ (the frequency ratio is $8 : 1$), and the region with large derivative magnitude is indeed the high frequency region. Moreover, Example 2 indicates that for one dimensional real-valued functions over a bounded domain, the integral of a derivative magnitude is closely related to the frequency information. For the squared derivative, the integral is the same as weighting the frequency terms $a_n$ and $b_n$ proportional to $n^2$, and for the squared second-order derivative, the integral is the same as weighting the frequency terms proportional to $n^4$. The weighting schemes $n^2$ or $n^4$ emphasize the higher frequency terms.

**Empirical demonstration.** Our calculation in the above examples implies that regions with large gradient and Hessian norm correspond to high frequency regions. Based on the same spirit of the $l_2$ regression task in Section 3.2, we empirically verify this insight. Our expectation is that biasing training dataset towards high gradient norm and Hessian norm would achieve better learning results. In Fig. 2(a), **Biased-GradientNorm** corresponds to uniformly sampling $x \in [-2, 2]$ for 60% of training data and sampling proportional to gradient norm (i.e., $p(x) \propto |f'_{\sin}(x)|$) for the remaining 40%; while **Biased-HessianNorm** corresponds to sampling proportional to Hessian norm (i.e., $p(x) \propto |f''_{\sin}(x)|$) for the remaining 40% of training data. In Fig. 2(b)(c), we visualize the two types of biased training points. Sampling according to the gradient norm or the Hessian norm leads to denser point distribution in the high frequency region $[-2, 0)$: there are $65.35\%, 68.97\%$ of training points fall in $[-2, 0]$ in Fig. 2(b), (c) respectively. An important difference between Fig. 2(b) and (c) is that, sampling according to Hessian norm leads to denser points around spikes: there are $18.17\%$ points fall in the yellow area in (b) and $27.45\%$ such points in (c). Those areas around spikes should be more difficult to approximate as the underlying function changes sharply, which explains the superior performance on the data set biased by Hessian norm. Fig. 2(a) shows that such biased training datasets provide fast learning, similar to the high frequency biased training datasets in Fig. 1.

Given a function $f : \mathcal{X} \mapsto \mathcal{Y}$ and a point $x \in \mathcal{X}$, we propose to measure frequency of $f$ around a small neighborhood of $x$ (we call this *local frequency*) using the following function:

$$g(x) \stackrel{\text{def}}{=} \|\nabla_x f(x)\|^2 + \|H_f(x)\|_F^2, \tag{2}$$

where $\|\nabla_x f(x)\|$ is the $\ell_2$-norm of the gradient at $x$, and $H_f(x)_F$ is the Frobenius norm of the Hessian matrix of $f$ at $x$. We claim that local frequency of $f$ around $x$ is proportional to $g(x)$. We theoretically justify this claim. For real-valued functions in Euclidean spaces, our theory connects local gradient and Hessian norms, local function energy [3], to local frequency distribution. The proof of our theorem and its connection to the well-known uncertainty principle are in Appendix A.2.

**Theorem 1.** *Given any function $f : \mathbb{R}^n \to \mathbb{R}$, for any frequency vector $k \in \mathbb{R}^n$, define its local Fourier transform around $x \in \mathbb{R}^n$,*

$$\hat{f}(k) \stackrel{\text{def}}{=} \int_{y \in B(x,1)} f(y) \exp\left\{-2\pi i \cdot y^\top k\right\} dy,$$

*for local function around $x$, i.e., $y \in B(x, 1) \stackrel{\text{def}}{=} \{y : \|y - x\| < 1\}$. Assume the local function "energy" is finite,*

$$\int_{y \in B(x,1)} [f(y)]^2 \, dy = \int_{\mathbb{R}^n} \|\hat{f}(k)\|^2 dk < \infty, \quad \forall x \in \mathbb{R}^n. \tag{3}$$

---

[3]We consider the notion of energy in signal processing terminology: the energy of a continuous time signal $x(t)$ is defined as $\int x(t)^2 dt$. In our theory, function $f$ is the signal.

*Define "local frequency distribution" of $f(x)$ as:*

$$\pi_{\hat{f}}(k) \stackrel{\text{def}}{=} \frac{\|\hat{f}(k)\|^2}{\int_{\mathbb{R}^n} \|\hat{f}(\tilde{k})\|^2 d\tilde{k}}, \quad \forall k \in \mathbb{R}^n. \tag{4}$$

*Then, for any $x \in \mathbb{R}^n$, we have:*
*1) The first order connection:*

$$\int_{y \in B(x,1)} \|\nabla f(y)\|^2 \, dy = 4\pi^2 \cdot \left[ \int_{y \in B(x,1)} [f(y)]^2 \, dy \right] \cdot \left[ \int_{\mathbb{R}^n} \pi_{\hat{f}}(k) \cdot \|k\|^2 \, dk \right], \tag{5}$$

*2) The second order connection:*

$$\int_{y \in B(x,1)} \|H_f(y)\|_F^2 dy = 16\pi^4 \left[ \int_{y \in B(x,1)} [f(y)]^2 \, dy \right] \cdot \left[ \int_{\mathbb{R}^n} \pi_{\hat{f}}(k) \cdot \|k\|^4 \, dk \right] \tag{6}$$

**Remark 1.** *Note that $\pi_{\hat{f}}$ defined in Eq. (4) is a probability distribution over $\mathbb{R}^n$ as:*

$$\int_{k \in \mathbb{R}^n} \pi_{\hat{f}}(k) dk = 1, \text{ and } \pi_{\hat{f}}(k) \geq 0, \quad \forall k \in \mathbb{R}^n.$$

*We use such a distribution to characterize local frequency behaviour for reasons. **First**, comparing frequencies of regions is more naturally captured by a distribution than one single scalar, since signals usually are within a range of frequencies. **Second**, to eliminate the impact of the function energy Eq. (3), we normalize the Fourier coefficient $\hat{f}$ to get $\pi_{\hat{f}}$.*

**Remark 2.** *For a frequency vector $k \in \mathbb{R}^n$, the larger its norm $\|k\|$ is, the higher its frequency is. Given any $x$ and its local function (i.e., $f(\cdot)$ around $x$), $\pi_{\hat{f}}(k)$ is the proportion/percentage that frequency $k$ occupies. Therefore, the integral of $\pi_{\hat{f}}(k) \cdot \|k\|^2$ reflects the contribution of high frequency terms in the local frequency distribution of a function.*

**Remark 3.** *Consider $f$ as a value function in reinforcement learning setting. Theorem 1 indicates that regions with large gradient norm can either have large absolute value function, or high local frequency, or both. To prevent finding regions that only have large negative value function, our theory implies that it is reasonable to take both gradient norm and value function into account, as our proposed method does in the next section.*

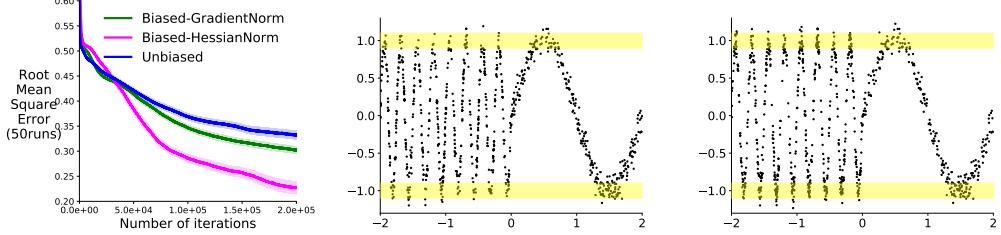

(a) RMSE vs. number of iterations  (b) biased training points - Gradient  (c) biased training points - Hessian

Figure 2: We show the learning curve of the $l_2$ regression on three training datasets in Figure (a) and show 1k points uniformly sampled from the two biased training data sets in (b)(c) respectively. The total number of training data points are the same across all experiments. The yellow area includes all the spikes, and is defined by restricting $||y| - 1.0| < 0.1$. The testing set is unbiased and the results are averaged over 50 random seeds with the shade indicating standard error.

## 4 FREQUENCY-BASED SEARCH-CONTROL IN DYNA

We present the Dyna architecture with the frequency-based search-control (Algorithm 1) in this section. It combines the idea that samples from high-frequency regions of the state space is important,

as discussed in the previous section, and the hill climbing process to effectively draw samples from those regions, as introduced by Pan et al. (2019). We omit implementation details such as preconditioning, noisy gradient for the hill climbing process, and refer readers to Appendix A.6 and A.7.

Our goal is to query the model more often from the regions of the state space where the local frequency of the value function is higher. The intuition behind this search-control mechanism, as discussed in the previous section in the context of supervised learning, is that those regions correspond to where learning the (value) function is more difficult, hence more samples from the model might be helpful. To populate the search-control queue with states from those regions, we can do hill climbing on $g(s) = \|\nabla_s V(s)\|^2 + \|H_v(s)\|_F^2$. Theorem 1, however, suggests that states with large gradient norm can either have large absolute value, or high local frequency, or both. We want to avoid many samples from regions with large negative value states, as those states may be rarely visited under the optimal policy anyway. A sensible strategy to get around this problem is to combine the proposed hill climbing method with the previous hill climbing on the value function (Pan et al., 2019), as the latter tends to generate samples from high value states.

We propose the following method for combining those approaches. At each time step, with certain probability, we perform hill climbing by either

$$s \leftarrow s + \alpha \begin{cases} \nabla_s g(s) & \text{with probability of } p \quad\quad (7a) \\ \nabla_s V(s) & \text{with probability of } 1 - p \quad\quad (7b) \end{cases}$$

and store states along the gradient trajectory in the search-control queue. When hill climbing on the value function (7b), we sample the initial state from the ER buffer as suggested by the previous work (Pan et al., 2019). This populates the search-control queue with states from the high value regions of the state space. When hill climbing on $g(s)$ (7a), however, we sample the initial state from the search-control queue itself (instead of the ER buffer). This way ensures that the initial state for searching high frequency region has relatively high value. Hill climbing on $g(s)$ from an initial state with a high value populates the search-control queue with high frequency samples around high value regions of the state space. We discuss some other intuitive mechanisms that we have tested in Appendix A.4.

Similar to the previous work (Pan et al., 2019), we obtain the state-value function in both (7a) and (7b) by taking the maximum of the estimated action-value, i.e. $V(s) = \max_a Q(s, a) \approx \max_a Q_\theta(s, a)$ where $\theta$ is the parameter of the $Q$-network. Similar to the Dyna architecture (Algorithm 2), during planning stage, we sample multiple mixed mini-batches to update the parameters (i.e. we call multiple planning steps/updates). The mixed mini-batch was also used in the work by Gu et al. (2016) and can alleviate off-policy sampling issue as studied by Pan et al. (2019).

## 5 EXPERIMENTS

In the experiments, we carefully study the properties of our algorithm on the MountainCar benchmark domain. Then we illustrate the utility of our algorithm on a challenging self-designed Maze-GridWorld domain, by which we illustrate the practical implication of having samples from the high frequency regions. Though we mainly focuses on search-control instead of how to learn a model, we include the result of using an online learned model for our algorithm. We refer readers to Appendix A.5 for additional experiments and Appendix A.6 for the reproducibility detail.

### 5.1 UTILITY OF FREQUENCY-BASED SEARCH-CONTROL

The MountainCar (Brockman et al., 2016) domain is well-studied, and it is known that the value function under the optimal value function has sharp changing regions (Sutton & Barto, 2018), which is the setting where our algorithm should be more effective. The agent needs to learn to reach the goal state within as few steps as possible since the reward is $-1$ per time step. The purposes of experimenting on this domain are: 1) verify that our search-control can outperform several natural competitors under different number of planning updates; 2) show that our search-control is robust to environment noise.

We use the following competitors. **Dyna-Frequency** is the Dyna architecture using the proposed search-control strategy (Algorithm 1); **Dyna-Value** is Algorithm 3 from the previous work (Pan et al., 2019); **PrioritizedER** is DQN with prioritized experience replay (Schaul et al., 2016); **ER**

---

**Algorithm 1** Dyna architecture with Frequency-based search-control

---

$B$: the ER buffer, $B_s$: search-control queue
$\mathcal{M} : \mathcal{S} \times \mathcal{A} \to \mathcal{S} \times \mathbb{R}$, the model outputs the next-state and reward
$m$: number of states we want to fetch by hill climbing, $d$: number of planning steps
$\epsilon_a$: the threshold for accepting a state
$Q, Q'$: current and target Q networks, respectively
$b$: the mini-batch size, $\beta \in (0, 1)$: the proportion of simulated samples in a mini-batch
$\tau$: update target network $Q'$ every $\tau$ updates to $Q$
$t \leftarrow 0$ is the time step, $n_\tau$ is the number of parameter updates
**while** true **do**
    Observe $s_t$, take action $a_t$ (i.e. $\epsilon$-greedy w.r.t. $Q$)
    Observe $s_{t+1}, r_{t+1}$, add $(s_t, a_t, s_{t+1}, r_{t+1})$ to $B$
    // Gradient ascent hill climbing
    With probability $p, 1 - p$, choose hill climbing rule (7a) or (7b) respectively;
    sample $s$ from $B_s$ if choose rule (7a), or from $B$ otherwise; set $c \leftarrow 0, \tilde{s} \leftarrow s$
    **while** $c < m$ **do**
        update $s$ by executing the chosen hill climbing rule
        **if** $s$ is out of state space **then**: // resample the initial state and hill climbing rule
            With probability $p, 1 - p$, choose hill climbing rule (7a) or (7b) respectively;
            sample $s$ from $B_s$ if choose (7a), or from $B$ otherwise; set $c \leftarrow 0, \tilde{s} \leftarrow s$
            **continue**
        **if** $||s - \tilde{s}||_2/\sqrt{n} > \epsilon_a$ **then**: // $n$ is the state dimension, i.e. $\mathcal{S} \subset \mathbb{R}^n$
            add $s$ to $B_s$, $\tilde{s} \leftarrow s, c \leftarrow c + 1$
    **for** $d$ times **do** // $d$ planning updates: sample $d$ mini-batches
        draw $\beta b$ sample states from the search-control queue $B_s$, pair them with their corresponding on-policy action, and query $\mathcal{M}$ to get the corresponding next-states and rewards
        draw $(1 - \beta)b$ sample transitions from the ER buffer $B$ and add them to the simulated transitions
        use the mixed mini-batch for parameter update of the estimator, e.g., DQN
        $n_\tau \leftarrow n_\tau + 1$
        **if** $\mod(n_\tau, \tau) == 0$ **then**:
            $Q' \leftarrow Q$
    $t \leftarrow t + 1$

---

is simply DQN with experience replay (ER) (Mnih et al., 2015). Figure 3 shows the learning curves of all those algorithms using 10 planning updates (a)(b) and 30 planning updates (c)(d) under different stochasticity. In Figure 3(b)(d), the reward is sampled from the Gaussian distribution $N(-1, \sigma^2), \sigma \in \{0.0, 0.1\}$.

We make several important observations: 1) With increased number of planning updates, these algorithms do not necessarily perform better, as shown in Figure 3(c). The proposed algorithm, however, appears to gain more through more number of updates since the difference between **Dyna-Frequency** and **Dyna-Value** seems to be clearer in Figure 3(c) than in Figure 3(a). 2) Since both **Dyna-Value** and our algorithm fetch the same number of states (i.e. $m = 20$) by hill climbing, the superior performance of our algorithm indicates the advantage of using samples from the high frequency regions. 3) **PrioritizedER** clearly performs worse than our algorithm and **Dyna-Value**, which probably implies the utility of the generalization power of the value function to acquire additional samples. 4) Our algorithm maintains superior performance in the presence of noise. One reason is that, noisy perturbation leads to more "energy" in all frequencies. When taking derivative, those high frequency terms are amplified. Hence, even with perturbation, high frequency region remains while the value estimate itself may get affected in an unpredictable manner.

## 5.2 A Case study: MazeGridWorld

We now illustrate the utility of our method on a challenging MazeGridWorld domain as shown in Figure 4(a). The domain has continuous state space $\mathcal{S} = [0, 1]^2$ and four discrete actions $\{\mathsf{up}, \mathsf{down}, \mathsf{left}, \mathsf{right}\}$. There are three walls in the middle, each of which has a hole for the agent

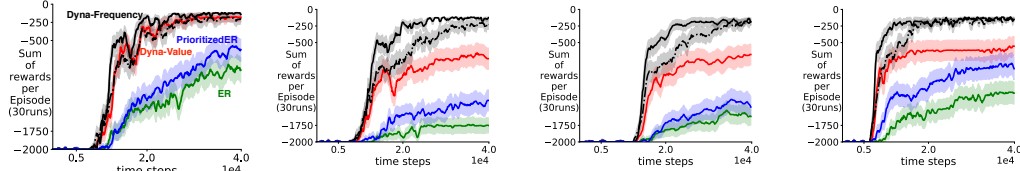

(a) plan steps 10, $\sigma = 0$    (b) plan steps 10, $\sigma = 0.1$    (c) plan steps 30, $\sigma = 0$    (d) plan steps 30, $\sigma = 0.1$

Figure 3: Evaluation curves (sum of episodic reward v.s. environment time steps) of **Dyna-Value**, **PrioritizedER**, **Dyna-Frequency**, **ER** on MountainCar with different number of planning updates with different reward noise variance. Notice that the **dashed** line denotes the evaluation curve of our algorithm with an online learned model. $\sigma = 0$ indicates the original deterministic reward. All results are averaged over 30 random seeds.

to go through. Each episode starts from the bottom left and ends at top right and the agent receives a reward of $-1$ at each time step, hence the agent should learn to use as few steps as possible to reach the goal area. On this domain, we mainly study our algorithm and the **Dyna-Value** algorithm.

Figure 4(b) shows the evaluation curves of the two algorithms. An important difference between our algorithm and the previous work is in the variance of the evaluation curve, which implies a robust policy learned by our method. In Figure 5, we further investigate the state distribution in search-control queues of the two algorithms by uniformly sampling 1000 states from the two queues. Notice that a very important difference between the two distributions is that our search-control queue has a clearly high density around the *bottleneck* area, i.e., the hole areas where the agent can go across the walls. Learning a stable policy around such areas is extremely important: the agent simply fails to reach the goal state if they cannot pass any one of the holes. This distinguishes our algorithm with the previous work, which appears to acquire states near the goal area.

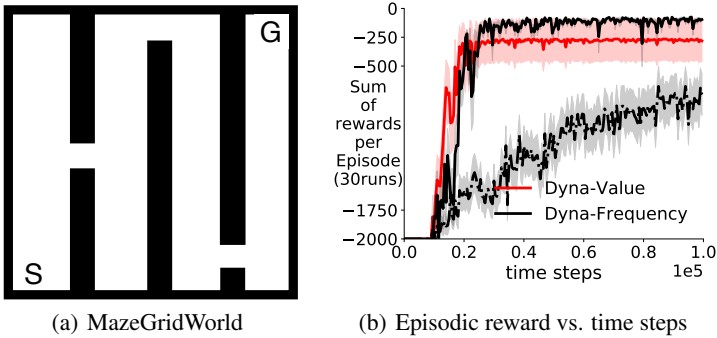

(a) MazeGridWorld      (b) Episodic reward vs. time steps

Figure 4: (a) is a visualization of the MazeGridWorld domain. (b) shows evaluation curves of **Dyna-Value** and **Dyna-Frequency**. **Dashed** line indicates using an online learned model of our algorithm. All results are averaged over 30 random seeds.

## 6   DISCUSSION

We motivated and studied a new category of methods for search-control by considering the approximation difficulty of a function. We provided a method for identifying the high frequency regions of a function, and justified it theoretically. We conducted experiments to illustrate our theory. We incorporated the proposed method into the Dyna architecture and empirically investigated its benefits. The method achieved competitive learning performances on a difficult domain.

There are several promising future research directions. First, it is worth exploring the combination of different search-control strategies. Second, we can explore the use of active learning methods (Settles, 2010; Hanneke, 2014) in the design of search-control mechanisms, since active learning con-

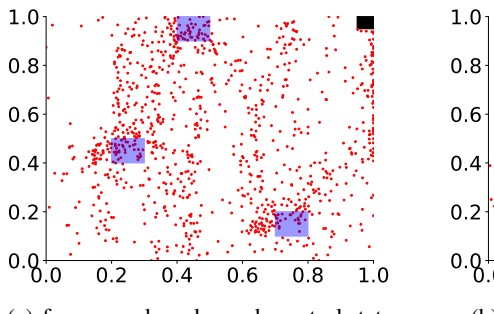
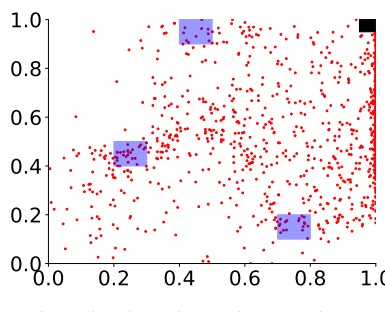

(a) frequency-based search-control states     (b) value-based search-control states

Figure 5: The state distribution in the search-control queue of our algorithm **Dyna-Frequency** (a) and **Dyna-Value** (b) at 50k environment time step. Each blue shadow area is a $0.1 \times 0.1$ square indicating the hole where the agent can go through the wall. Our search-control queue has a state distribution with a high density around those squares. In (a), there are $25.3\%$ points fall inside a $0.1$ radius ball centered at each square in total; in (b), there are $11.7\%$ such points. The black box on the top right is the goal area.

cerns about learning a function with as few samples as possible. This directly corresponds to our main purpose of using a smart search-control method in Dyna: to improve policy by using as few planning steps as possible.

ACKNOWLEDGMENTS

We would like to thank the anonymous reviewers for their helpful feedback. Amir-massoud Farahmand acknowledges the funding from the the Canada CIFAR AI Chairs program. Yangchen Pan acknowledges the funding from Amii.

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

## A  APPENDIX

We provide calculations for Example 1 and Example 2 in Section A.1; theoretical proof of Theorem 1 in Section A.2. Background of Dyna is reviewed in Section A.3 and additional discussions regarding search-control are provided in Section A.4. Additional experiments regarding separate criterion of hill climbing, and on continuous control problems are shown in Section A.5. Experimental details for reproducing our empirical results are in Section A.6.

### A.1 CALCULATIONS FOR EXAMPLE 1 AND EXAMPLE 2

**Example 1.** For $f_{\sin}$ defined in Eq. (1), calculate the integrals of squared first order derivative $f'_{\sin}$ on high frequency region $[-2, 0)$ and low frequency region $[0, 2]$, respectively:

$$\int_{-2}^{0} [f'_{\sin}(x)]^2 \, dx = 64\pi^2, \quad \int_{0}^{2} [f'_{\sin}(x)]^2 \, dx = \pi^2.$$

*Proof.* Taking derivative and integral,

$$\int_{-2}^{0} [f'(x)]^2 \, dx = 64\pi^2 \int_{-2}^{0} [\cos(8\pi x)]^2 \, dx = 64\pi^2,$$

$$\int_{0}^{2} [f'(x)]^2 \, dx = \pi^2 \int_{0}^{2} [\cos(\pi x)]^2 \, dx = \pi^2. \qquad \square$$

**Example 2.** Let $f : [-\pi, \pi] \to \mathbb{R}$ be a band-limited real valued function defined as

$$f(x) = \frac{a_0}{2} + \sum_{n=1}^{N} a_n \cos(nx) + b_n \sin(nx),$$

where $a_0, a_n, b_n \in \mathbb{R}$, $n = 1, 2, \ldots, N$ are Fourier coefficients of frequency $\frac{n}{2\pi}$. Then,

$$\int_{-\pi}^{\pi} |f'(x)|^2 \, dx = \pi \cdot \sum_{n=1}^{N} n^2 \left(a_n^2 + b_n^2\right), \quad \int_{-\pi}^{\pi} |f''(x)|^2 \, dx = \pi \cdot \sum_{n=1}^{N} n^4 \left(a_n^2 + b_n^2\right).$$

*Proof.* Taking derivative of $f$,

$$f'(x) = \sum_{n=1}^{N} [-na_n \sin(nx)] + \sum_{n=1}^{N} [nb_n \cos(nx)].$$

Taking square of $f'$,

$$[f'(x)]^2 = \sum_{n=1}^{N} \sum_{m=1}^{N} [nma_n a_m \sin(nx) \sin(mx)] - \sum_{n=1}^{N} \sum_{m=1}^{N} [nma_n b_m \sin(nx) \cos(mx)]$$

$$- \sum_{n=1}^{N} \sum_{m=1}^{N} [mna_m b_n \sin(mx) \cos(nx)] + \sum_{n=1}^{N} \sum_{m=1}^{N} [nmb_n b_m \cos(nx) \cos(mx)].$$

Taking integral,

$$\int_{-\pi}^{\pi} [f'(x)]^2 \, dx = \int_{-\pi}^{\pi} \sum_{n=1}^{N} \sum_{m=1}^{N} [nma_n a_m \sin(nx) \sin(mx)] dx - \int_{-\pi}^{\pi} \sum_{n=1}^{N} \sum_{m=1}^{N} [nma_n b_m \sin(nx) \cos(mx)] dx$$

$$- \int_{-\pi}^{\pi} \sum_{n=1}^{N} \sum_{m=1}^{N} [mna_m b_n \sin(mx) \cos(nx)] dx + \int_{-\pi}^{\pi} \sum_{n=1}^{N} \sum_{m=1}^{N} [nmb_n b_m \cos(nx) \cos(mx)] dx$$

$$= \sum_{n=1}^{N} \sum_{m=1}^{N} [nma_n a_m \pi \delta_{n,m} - 0 - 0 + nmb_n b_m \pi \delta_{n,m}]$$

$$= \pi \cdot \sum_{n=1}^{N} n^2 \left(a_n^2 + b_n^2\right),$$

where

$$\delta_{n,m} \stackrel{\text{def}}{=} \begin{cases} 1, & \text{if } n = m, \\ 0, & \text{otherwise.} \end{cases}$$

Using similar arguments, taking derivative of $f'(x)$,

$$f''(x) = \sum_{n=1}^{N} \left[ -n^2 a_n \cos(nx) \right] + \sum_{n=1}^{N} \left[ -n^2 b_n \sin(nx) \right].$$

Taking integral,

$$\int_{-\pi}^{\pi} \left[ f''(x) \right]^2 dx = \pi \cdot \sum_{n=1}^{N} n^4 \left( a_n^2 + b_n^2 \right). \qquad \square$$

## A.2 PROOF FOR THEOREM 1

**Notations.** For any vector norm $|| \cdot ||$, we mean $l_2$ norm and we ignore the subscript unless clarification is needed. We use Frobenius norm $|| \cdot ||_F$ for matrix. We use subscript $y_l$ to denote the $l$th element in vector $y$. Let $H_f(y)$ be the Hessian matrix of $f(y)$. We write $H$ for short unless clarification is needed. Let $H_{l,:}$ be the $l$th row of the Hessian matrix.

**Proof description.** We establish the connection between local gradient norm, Hessian norm and local frequency. To build such connection, we introduce a definition of $\pi_{\hat{f}}$ as shown below and we call it "local frequency distribution" of $f(x)$. $\pi_{\hat{f}}$ is a probability distribution over $\mathbb{R}^n$, i.e., $\int_{k \in \mathbb{R}^n} \pi_{\hat{f}}(k) dk = 1$, and $\pi_{\hat{f}}(k) \geq 0$, $\forall k \in \mathbb{R}^n$. Within an open subset of domain (an unit ball), this distribution characterizes the proportion of a particular frequency component occupies. The proof can be described by three key steps:1) We use a local Fourier transform to express a function locally (i.e. within an unit ball). 2) we calculate the gradient/Hessian norm based on this local Fourier transform; 3) we take integration over the unit ball of the gradient/Hessian norm to build the connection with the local frequency distribution $\pi_{\hat{f}}$ and function energy.

**Theorem 1.** Given any function $f : \mathbb{R}^n \to \mathbb{R}$, for any frequency vector $k \in \mathbb{R}^n$, define its local Fourier transform of $x \in \mathbb{R}^n$,

$$\hat{f}(k) \stackrel{\text{def}}{=} \int_{y \in B(x,1)} f(y) \exp \left\{ -2\pi i \cdot y^\top k \right\} dy,$$

for local function around $x$, i.e., $y \in B(x,1) \stackrel{\text{def}}{=} \{y : ||y - x|| < 1\}$. Assume the local function "energy" is finite,

$$\int_{y \in B(x,1)} \left[ f(y) \right]^2 dy = \int_{\mathbb{R}^n} ||\hat{f}(k)||^2 dk < \infty, \quad \forall x \in \mathbb{R}^n.$$

Define "local frequency distribution" of $f(x)$ as:

$$\pi_{\hat{f}}(k) \stackrel{\text{def}}{=} \frac{||\hat{f}(k)||^2}{\int_{\mathbb{R}^n} ||\hat{f}(\tilde{k})||^2 d\tilde{k}}, \quad \forall k \in \mathbb{R}^n.$$

Then, $\forall x \in \mathbb{R}^n$, we have:
1) the first order connection:

$$\int_{y \in B(x,1)} ||\nabla f(y)||^2 dy = 4\pi^2 \cdot \left[ \int_{y \in B(x,1)} \left[ f(y) \right]^2 dy \right] \cdot \left[ \int_{\mathbb{R}^n} \pi_{\hat{f}}(k) \cdot ||k||^2 dk \right],$$

2) the second order connection:

$$\int_{y \in B(x,1)} ||H(y)||_F^2 dy = 16\pi^4 \left[ \int_{y \in B(x,1)} \left[ f(y) \right]^2 dy \right] \cdot \left[ \int_{\mathbb{R}^n} \pi_{\hat{f}}(k) \cdot ||k||^4 dk \right]$$

*Proof.* 1) We first prove the first order connection.
Consider the following function defined locally around $x$,

$$f_x(y) \stackrel{\text{def}}{=} \begin{cases} f(y), & \text{if } y \in B(x,1), \\ 0, & \text{otherwise.} \end{cases}$$

By definition, the Fourier transform of $f_x$ is

$$\hat{f}(k) = \int_{y \in B(x,1)} f(y) \exp\left\{-2\pi i \cdot y^\top k\right\} dy$$

$$= \int_{\mathbb{R}^n} f_x(y) \exp\left\{-2\pi i \cdot y^\top k\right\} dy.$$

And the inverse Fourier transform of $f_x(y), \forall y \in B(x,1)$ is,

$$f_x(y) = \int_{\mathbb{R}^n} \hat{f}(k) \exp\left\{2\pi i \cdot y^\top k\right\} dk,$$

and then the gradient $\forall y \in B(x,1)$ is

$$\nabla f(y) = \nabla f_x(y) = \int_{\mathbb{R}^n} \hat{f}(k) \exp\left\{2\pi i \cdot y^\top k\right\} (2\pi i \cdot k) \, dk. \tag{8}$$

To calculate gradient norm, we use complex conjugate,

$$\nabla f^*(y) = \int_{\mathbb{R}^n} \hat{f}^*(k') \exp\left\{-2\pi i \cdot y^\top k'\right\} (-2\pi i \cdot k') \, dk',$$

where

$$\hat{f}^*(k') = \int_{\mathbb{R}^n} f_x(y') \exp\left\{2\pi i \cdot {y'}^\top k'\right\} dy'$$

is the complex conjugate of $\hat{f}(k)$. Therefore,

$$\begin{aligned}
\|\nabla f(y)\|^2 &= \langle \nabla f(y), \nabla f^*(y) \rangle \\
&= \int_{\mathbb{R}^n} \int_{\mathbb{R}^n} \hat{f}(k)\hat{f}^*(k') \exp\left\{2\pi i \cdot y^\top (k - k')\right\} \left(4\pi^2 k^\top k'\right) dk dk'.
\end{aligned} \tag{9}$$

Taking integral of $\|\nabla f(y)\|^2$ within the unit ball centered at $x$,

$$\int_{y \in B(x,1)} \|\nabla f(y)\|^2 \, dy = \int_{\mathbb{R}^n} \|\nabla f_x(y)\|^2 \, dy, \text{ by function definition} \tag{10a}$$

$$= \int_{\mathbb{R}^n} \int_{\mathbb{R}^n} \hat{f}(k)\hat{f}^*(k') \left[\int_{\mathbb{R}^n} \exp\left\{2\pi i \cdot y^\top (k - k')\right\} dy\right] \left(4\pi^2 k^\top k'\right) dk dk' \tag{10b}$$

$$= \int_{\mathbb{R}^n} \int_{\mathbb{R}^n} \hat{f}(k)\hat{f}^*(k')\delta_{k-k',\mathbf{0}} \left(4\pi^2 k^\top k'\right) dk dk' \tag{10c}$$

$$= 4\pi^2 \int_{\mathbb{R}^n} \|\hat{f}(k)\|^2 \cdot \|k\|^2 \, dk. \tag{10d}$$

Recall the definition of local function "energy" around $x$,

$$\int_{\mathbb{R}^n} \|\hat{f}(k)\|^2 dk = \int_{\mathbb{R}^n} \left\langle \hat{f}(k), \hat{f}^*(k) \right\rangle dk \tag{11a}$$

$$= \int_{y \in \mathbb{R}^n} \int_{y \in \mathbb{R}^n} f_x(y)f_x(y') \left[\int_{\mathbb{R}^n} \exp\left\{2\pi i \cdot k^\top (y' - y)\right\} dk\right] dy dy' \tag{11b}$$

$$= \int_{y \in \mathbb{R}^n} \int_{y \in \mathbb{R}^n} f_x(y)f_x(y')\delta_{y'-y,\mathbf{0}} dy dy' \tag{11c}$$

$$= \int_{y \in \mathbb{R}^n} f_x^2(y) dy \tag{11d}$$

$$= \int_{y \in B(x,1)} f^2(y) dy < \infty. \quad \text{(by definition of } f_x(y) \text{ and finite energy assumption)} \tag{11e}$$

For $y \in B(x,1)$, the local gradient information is related to local energy and frequency distribution,

$$\int_{y \in B(x,1)} \|\nabla f(y)\|^2 \, dy = 4\pi^2 \int_{\mathbb{R}^n} \|\hat{f}(k)\|^2 \cdot \|k\|^2 \frac{\int_{\mathbb{R}^n} \|\hat{f}(\tilde{k})\|^2 d\tilde{k}}{\int_{\mathbb{R}^n} \|\hat{f}(\tilde{k})\|^2 d\tilde{k}} dk \tag{12a}$$

$$= 4\pi^2 \int_{\mathbb{R}^n} \pi_{\hat{f}}(k) \, \|k\|^2 \int_{\mathbb{R}^n} \|\hat{f}(\tilde{k})\|^2 d\tilde{k} dk \tag{12b}$$

$$= 4\pi^2 \cdot \left[ \int_{y \in B(x,1)} f^2(y) dy \right] \cdot \left[ \int_{\mathbb{R}^n} \pi_{\hat{f}}(k) \cdot \|k\|^2 \, dk \right], \tag{12c}$$

where the last equality follows by $\int_{\mathbb{R}^n} \|\hat{f}(\tilde{k})\|^2 d\tilde{k} = \int_{y \in B(x,1)} f^2(y) dy$ which is established in the derivation (11).

2) Now we prove the second order connection.
To show the second order connection, we start from Eq. (8). Then the $l$th row of the Hessian matrix $H_{l,:}$ can be written as:

$$H_{l,:} = \frac{\partial \nabla f(y)}{\partial y_l}^{\top}$$

where we use the notation $\frac{\partial \nabla f(y)}{\partial y_l}$ to denote the vector formed by taking partial derivative of each element in the gradient vector $\nabla f(y)$ w.r.t. $y_l$. Then,

$$\frac{\partial \nabla f(y)}{\partial y_l} = \int_{\mathbb{R}^n} \hat{f}(k) \exp\left\{2\pi i \cdot y^{\top} k\right\} \left(4\pi^2 i^2 (e_l^{\top} k) k\right) dk,$$

where $e_l$ is standard basis vector where the $l$th element is one. To calculate the norm of the vector $H_{l,:} = \frac{\partial \nabla f(y)}{\partial y_l}^{\top}$, we use complex conjugate again and follow the similar derivation as done in Eq. (9):

$$\|H_{l,:}\|_2^2 = \langle H_{l,:}, H_{l,:} \rangle$$

$$= \int_{\mathbb{R}^n} \int_{\mathbb{R}^n} \hat{f}(k) \hat{f}^*(k') \exp\left\{2\pi i \cdot y^{\top}(k - k')\right\} \left(16\pi^4 i^4 (e_l^{\top} k)(e_l^{\top} k') k^{\top} k'\right) dk dk'$$

Note that the square of Frobenius norm of the Hessian matrix can be written as $\|H\|_F^2 = \sum_{i,j} H_{i,j}^2 = \sum_{l=1}^{n} \|H_{l,:}\|_2^2$. Then,

$$\|H\|_F^2 = \sum_{l=1}^{n} \|H_{l,:}\|_2^2$$

$$= \int_{\mathbb{R}^n} \int_{\mathbb{R}^n} \hat{f}(k) \hat{f}^*(k') \exp\left\{2\pi i \cdot y^{\top}(k - k')\right\} \left(16\pi^4 i^4 \sum_{l=1}^{n} (e_l^{\top} k)(e_l^{\top} k') k^{\top} k'\right) dk dk'$$

$$= 16\pi^4 \int_{\mathbb{R}^n} \int_{\mathbb{R}^n} \hat{f}(k) \hat{f}^*(k') \exp\left\{2\pi i \cdot y^{\top}(k - k')\right\} (k^{\top} k')^2 dk dk'$$

Taking the integration of $\|H\|_F^2$ over $y$ variable within a ball with center $x$ and unit radius, we acquire:

$$\int_{y \in B(x,1)} \|H(y)\|_F^2 dy = 16\pi^4 \int_{\mathbb{R}^n} \|\hat{f}(k)\|^2 \|k\|^4 dk$$

$$= 16\pi^4 \left[ \int_{y \in B(x,1)} [f(y)]^2 \, dy \right] \cdot \left[ \int_{\mathbb{R}^n} \pi_{\hat{f}}(k) \cdot \|k\|^4 \, dk \right]$$

where the derivation process for the first equation is a simple modification from the derivation (10) and the second equation follows the same derivation (12). $\qquad \square$

---

**Algorithm 2** Generic Dyna Architecture: Tabular Setting

---

Initialize $Q(s,a)$ and model $\mathcal{M}(s,a), \forall (s,a) \in \mathcal{S} \times \mathcal{A}$
**while** true **do**
    observe $s$, take action $a$ by $\epsilon$-greedy w.r.t $Q(s,\cdot)$
    execute $a$, observe reward $R$ and next state $s'$
    Q-learning update for $Q(s,a)$
    update model $\mathcal{M}(s,a)$ (i.e. by counting)
    store $(s,a)$ into search-control queue
    **for** i=1:d **do**
        sample $(\tilde{s},\tilde{a})$ from search-control queue
        $(\tilde{s}',\tilde{R}) \leftarrow \mathcal{M}(\tilde{s},\tilde{a})$ // simulated transition
        Q-learning update for $Q(\tilde{s},\tilde{a})$ // planning update

---

**Discussion with Uncertainty Principle.** We now provide an intuitive interpretation of our theorem from the well-known uncertainty principle. The Uncertainty Principle says that a function cannot be simultaneously concentrated in both spatial and frequency space. That is, the more concentrated a function is, the more spread out its Fourier transform must be, indicating that more high frequency terms are needed to express the function. For example, by uncertainty principle (Stein & Shakarchi, 2003), $\forall x \in \mathbb{R}$ we have

$$\left[ \int_{\|y-x\|\leq 1} (y-x)^2 \cdot [f_x(y)]^2 \, dy \right] \cdot \left[ \int_{\mathbb{R}^n} \|\hat{f}(k)\|^2 \cdot \|k\|^2 \, dk \right] \geq \frac{1}{16\pi^2}. \tag{13}$$

Note that the first term $\left[ \int_{\|y-x\|\leq 1} (y-x)^2 \cdot [f_x(y)]^2 \, dy \right]$ measures the dispersion of the function around the point $x$. Hence the smaller this term is, the more concentration of the function is on the specified domain. Plug into our Eq. (5) and replace the function energy term by Fourier transform and cancel out the normalization term, we get:

$$\left[ \int_{\|y-x\|\leq 1} (y-x)^2 \cdot [f_x(y)]^2 \, dy \right] \cdot \left[ \int_{\|y-x\|\leq 1} \|\nabla f(y)\|^2 \, dy \right] \geq \frac{1}{4},$$

which means the more concentrated $f$ is locally around $x$, the larger the lower bound of the local gradient norm must be.

## A.3 BACKGROUND IN DYNA

In this section, we provide the vanilla (tabular) Dyna (Sutton, 1991a; Sutton & Barto, 2018) in Algorithm 2, and Hill Climbing Dyna by Pan et al. (2019) in Algorithm 3. Dyna is a classic model-based reinforcement learning architecture. As described in Algorithm 2, at each time step, the real experience is used to directly improve policy/value estimates, and is also used to learn the environment dynamics model. During planning stage, simulated experiences are acquired from the learned model and are used to further improve the policy. The critical component in Dyna is the search-control mechanism, which decides what simulated experiences to use during planning. This area is relatively unexplored, though abundant literature is available regarding how to learn a model.

## A.4 A DISCUSSION ON SEARCH-CONTROL DESIGN BASED ON HILL CLIMBING

There are different ways to combine hill climbing strategies. Here are some unsuccessful trials. For example, climbing on direct combinations of $V(s)$ (value function) and $g(s)$ (frequency criterion), such as $V(s) + g(s)$, or $V(s)g(s)$, did not work well. The reasons are as following. First, such combination can lead to unpredictable gradient behaviour. It can alter the trajectory solely based on either $g(s)$ or $V(s)$, and the effect is unclear. It may lead to states with neither high value or high frequency. Last, and probably the most important, hill climbing on $V(s)$ and on $g(s)$ have fundamentally different insights. The former is based on the intuition that the value information should be propagated from the high value region to low value region; as a result, it requires to store states along the whole trajectory, including those in low value region. This is empirically verified by Pan et al. (2019). However, the latter is based on the insight that the function value

---

**Algorithm 3** HC-Dyna architecture

---

$B_s$: search-control queue, $B$: the experience replay buffer
$m$: number of states to fetch by search-control
$b$: the mini-batch size
**while** true **do**
    Observe $s_t$, take action $a_t$ (i.e. $\epsilon$-greedy w.r.t. action value function)
    Observe $s_{t+1}, r_{t+1}$, add $(s_t, a_t, s_{t+1}, r_{t+1})$ to $B$
    sample $s$ from visited states, i.e. ER buffer $B$
    // Hill climbing by gradient ascent
    **while** get less than $m$ states **do**
        $s \leftarrow s + \nabla_s V(s), V(s) = \max_s Q_\theta(s, a)$
        store $s$ into search control queue $B_s$
    // Planning stage
    **for** $d$ times **do**
        // sample states from $B_s$ and pair them with on-policy actions, query the model to get next states and rewards
        // mix simulated and real experiences into a mini-batch and use it to update parameters (i.e. DQN update)
    $t \leftarrow t + 1$

---

in high frequency region is more difficult to approximate and needs more samples, while there is no obvious reason to propagate those information back to low frequency region. As a result, this approach does not emphasize on recording states throughout the whole hill climbing trajectory.

## A.5 ADDITIONAL EXPERIMENTS

In this section, we briefly study the effect of doing hill climbing on only gradient norm or Hessian norm. Then we demonstrate that our search-control strategy can be also used for continuous control algorithms.

**Hill climbing on only gradient norm or Hessian norm.** Throughout our paper, we use the form of $g(s) = \|\nabla_s V(s)\|^2 + \|H_v(s)\|_F^2$ to search states from high (local) frequency region of the value function. Besides the theoretical reason, there is a practical demand of such design. On value function surface, regions which have low (or even zero) gradient magnitude may have high Hessian magnitude, and vice versa. Hence, it can help move along the gradient trajectory in case that one of the term vanished at some point. Such cases can be a result of function approximation (smoothness/differentiability), or of the nature of the task, or both. In Fig. 6, we show the results of using only either gradient norm or Hessian norm. The reason we choose MountainCar and GridWorld (the same domain as described by Pan et al. (2019)) is that, the former has a value function surface with lots of variations; while the latter's value function increases smoothly from the initial state to the goal state, which indicates a small magnitude second-order derivative. Indeed, we empirically observe that the term $\nabla_s\|H_v(s)\|_F^2$ frequently gives a zero vector on GridWorld. This explains the bad performance of Dyna-HessNorm in Fig. 6(b). In contrast, Fig. 6(a) shows slightly better performance of Dyna-HessNorm and Dyna-GradNorm. Notice that, an intuitive and more general form of $g(x)$ can be $g(s) = \eta_1\|\nabla_s V(s)\|^2 + \eta_2\|H_v(s)\|_F^2$, at the cost that additional meta-parameters are introduced.

**Continuous Control.** In this section, we show a simple demonstration where our method is adapted to two continuous control tasks: Hopper-v2 and Walker2d-v2 from Mujoco (Todorov et al., 2012) by using a continuous $Q$ learning algorithm called NAF (Normalized Advantage Function) (Gu et al., 2016). The algorithm parameterizes the action value function as $Q(s, a) = V(s) - (a - \mu(s))^T P(a - \mu(s))$ where $P$ is a positive semi-definite matrix and hence the action with maximum value can be easily found: $\arg\max_a Q(s, a) = \mu(s)$. Our search-control strategy naturally applies here by utilizing the value function $V(s)$. From Fig. 7, one can see that our algorithm (**DynaNAF-Frequency**) finds a better policy comparing with the model-free NAF.

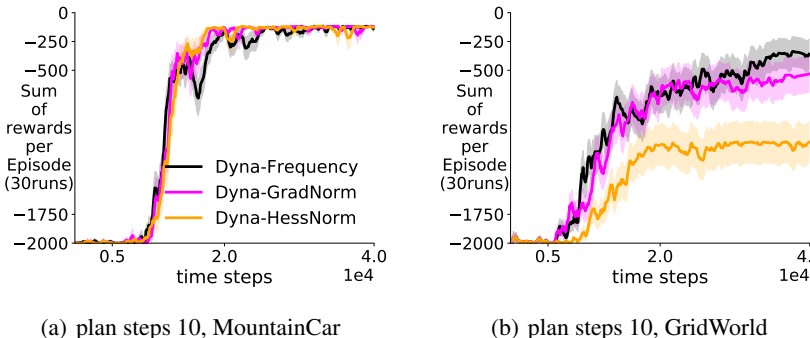

(a) plan steps 10, MountainCar   (b) plan steps 10, GridWorld

Figure 6: Evaluation curves (sum of episodic reward v.s. environment time steps) of hill climbing on gradient norm (Dyna-GradNorm) and Hessian norm (Dyna-HessNorm) on MountainCar and GridWorld with 10 planning updates. All results are averaged over 30 random seeds.

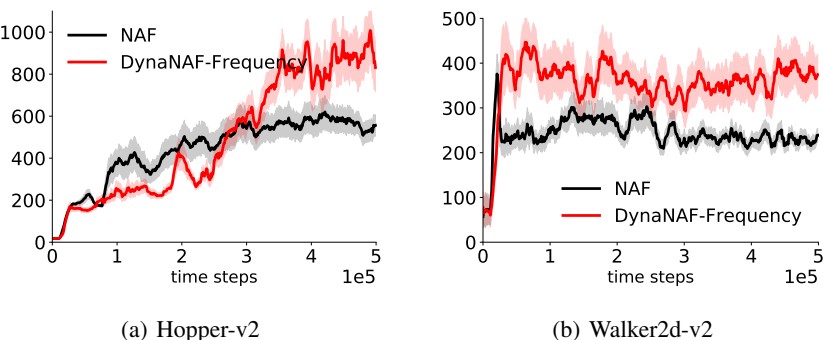

(a) Hopper-v2   (b) Walker2d-v2

Figure 7: The learning curves showing sum of rewards per episode as a function of environment time steps. We use 5 planning steps for both algorithm. The results are averaged over 10 random seeds.

## A.6 REPRODUCIBILITY DETAIL

All of our implementations are based on tensorflow with version 1.13.0 (Abadi et al., 2015). For DQN update, we use Adam optimizer (Kingma & Ba, 2014). We use mini-batch size $b = 32$ except on the supervised learning experiment where we use 128. For reinforcement learning experiment, we use buffer size 100k. All activation functions are tanh except the output layer of the $Q$-value is linear. Except the output layer parameters which were initialized from a uniform distribution $[-0.003, 0.003]$, all other parameters are initialized using Xavier initialization (Glorot & Bengio, 2010). For model learning, we use a $64 \times 64$ relu units neural network to predict $s' - s$ given a state-action pair with mini-batch size 128 and learning rate 0.0001.

For the supervised learning experiment shown in Section 3, we use $16 \times 16$ tanh units neural network, with learning rate 0.001 for all algorithms. The learning curve is plotted by computing the testing error every 20 iterations. When generating Fig. 2, in order to sample points according to $p(x) \propto |f'(x)|$ or $p(x) \propto |f''(x)|$, we use $10,000$ even spaced points on the domain $[-2, 2]$ and the probabilities are computed by normalization across the 10k points.

The experiment on MountainCar is based on the implementation from OpenAI (Brockman et al., 2016), we use $32 \times 32$ tanh layer, with target network moving rate 1000 and learning rate 0.001. Exploration noise is 0.1 without decaying. For all algorithms, we use warm up steps = 5000 (i.e. random action is taken in the first 5k time steps). Prioritized experience replay (PrioritizedER) is implemented as the proportional version with sum tree data structure. We use prioritized ER without importance ratio but half of mini-batch samples are uniformly sampled from the buffer as a strategy for bias correction. For Dyna-Value and Dyna-Frequency, we use: gradient ascent step size (in

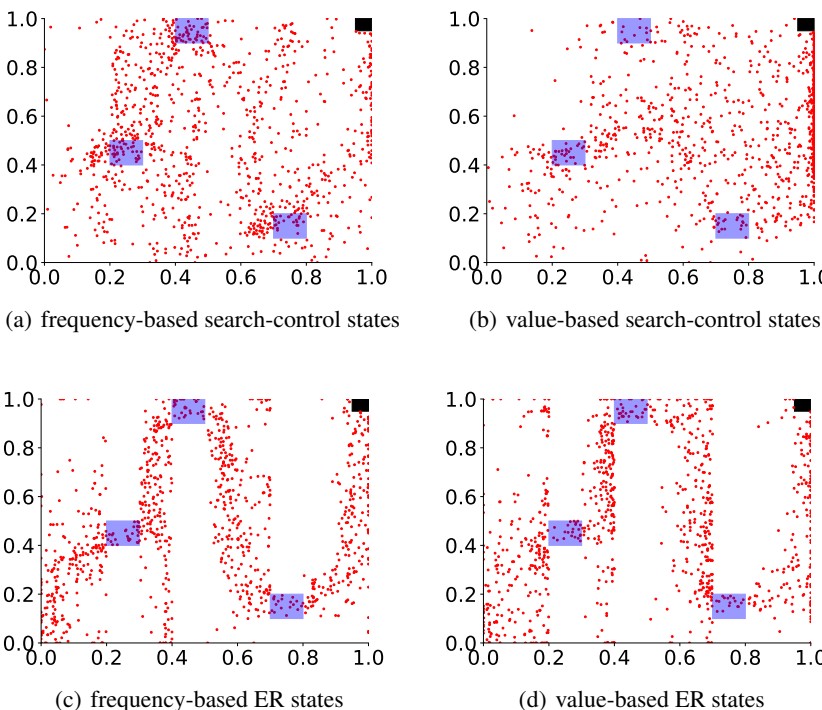

Figure 8: The state distribution in the search-control queue and ER buffer at $50,000$ environment time step. The blue shadow indicates the hole area where the agent can go through the wall. The black box on the right top is the goal area.

search-control) $0.01$, mixing rate $\beta = 0.5$ and $m = 20$, i.e., at each environment time step we fetch 20 states by hill climbing. We fix $p = 0.5$ across all experiments, hence the hill climbing rules (7a) and (7b) are selected with equal probability. We use natural projected gradient ascent for hill climbing as introduced by Pan et al. (2019).

For the experiment on MazeGridWorld, each wall's width is $0.1$ and each hole has height $0.1$. The left-top point of the hole in the first wall (counting from left to right) has coordinate $(0.2, 0.5)$; the hole in the second wall has coordinate $(0.4, 1.0)$ and the third one is $0.7, 0.2$. Each action leads to $0.05$ unit move perturbed by a Gaussian noise from $N(0, 0.01)$. On this domain, for both Dyna-Value and Dyna-Frequency, all parameters are set the same with that used on MountainCar except that we use $64 \times 64$ tanh units for $Q$ network, and number of search-control samples is set as $m = 50$, number of planning updates is 30. As a supplement to the Section 5.2, we also provide the state distribution from ER buffer in Figure 8. One can see that ER buffer has very different state distribution with search-control queue.

## A.7 ALGORITHMIC DETAILS

We provide the pseudo-code in Algorithm 4 with sufficient details to recreate our experimental results. The hill climbing rules we used is the same as introduced by Pan et al. (2019). Define

$$v_s \stackrel{\text{def}}{=} \nabla_s \max_a Q(s, a),$$

then

$$g_s \stackrel{\text{def}}{=} \nabla_s g(s) = \nabla_s(||\nabla_s \max_a Q(s, a)||_2^2 + ||H_v(s)||_F^2) = \nabla_s(||v_s||_2^2 + ||\nabla_s v_s||_F^2).$$

Note that we use a squared norm to ensure numerical stability when taking gradient. Then for value-based search-control, we use

$$s \leftarrow s + \frac{\alpha}{||\hat{\Sigma}_s v_s||}\hat{\Sigma}_s v_s + X_i, X_i \sim N(0, \eta\hat{\Sigma}_s) \tag{14}$$

and for frequency-based search-control, we use

$$s \leftarrow s + \frac{\alpha}{||\hat{\Sigma}_s g_s||} \hat{\Sigma}_s g_s + X_i, X_i \sim N(0, \eta \hat{\Sigma}_s) \tag{15}$$

where $\hat{\Sigma}_s$ is empirical covariance matrix estimated from visited states, and we set $\eta = 0.01, \alpha = 0.01$ across all experiments. Notice that comparing with the previous work, we omitted the projection step as we found it is unnecessary in our experiments.

---

**Algorithm 4** Dyna architecture with Frequency-based search-control with additional details

---

$B_s$: search-control queue, $B$: the experience replay buffer
$\mathcal{M} : \mathcal{S} \times \mathcal{A} \rightarrow \mathcal{S} \times \mathbb{R}$, the environment model
$m$: number of search-control samples to fetch at each step
$p$: probability of choosing value-based hill climbing rule (we set $p = 0.5$ for all experiments)
$\beta \in [0, 1]$: mixing factor in a mini-batch, i.e. $\beta b$ samples in a mini-batch are simulated from model
$n$: number of state variables, i.e. $\mathcal{S} \subset \mathbb{R}^n$
$\epsilon_a$: empirically learned threshold as sample average of $||s_{t+1} - s_t||_2/\sqrt{n}$
$d$: number of planning steps
$Q, Q'$: current and target Q networks, respectively
$b$: the mini-batch size
$\tau$: update target network $Q'$ every $\tau$ updates to $Q$
$t \leftarrow 0$ is the time step
$n_\tau \leftarrow 0$ is the number of parameter updates
// Gradient ascent hill climbing
With probability $p, 1 - p$, choose hill climbing Eq. (14) o Eq. (15) respectively;
sample $s$ from $B_s$ if choose rule Eq. (14), or from $B$ otherwise; set $c \leftarrow 0, \tilde{s} \leftarrow s$
**while** $c < m$ **do**
    update $s$ by executing the chosen hill climbing rule
    **if** $s$ is out of state space **then**: // resample the initial state and hill climbing rule
        With probability $p, 1 - p$, choose hill climbing rule Eq. (14) or Eq. (15) respectively;
        sample $s$ from $B_s$ if choose Eq. (7), or from $B$ otherwise; set $c \leftarrow 0, \tilde{s} \leftarrow s$
        **continue**
    **if** $||s - \tilde{s}||_2/\sqrt{n} > \epsilon_a$ **then**:
        add $s$ to $B_s$, $\tilde{s} \leftarrow s, c \leftarrow c + 1$
// $d$ planning updates: sample $d$ mini-batches
**for** $d$ times **do** // $d$ planning updates
    sample $\beta b$ states from $B_s$ and pair them with on-policy actions, and query $\mathcal{M}$ to get next states and rewards
    sample $b(1 - \beta)$ transitions from $B$ an stack these with the simulated transitions
    use the mixed mini-batch for parameter (i.e. DQN) update
    $n_\tau \leftarrow n_\tau + 1$
    **if** $mod(n_\tau, \tau) == 0$ **then**:
        $Q' \leftarrow Q$
$t \leftarrow t + 1$

---

