# OpenReview forum: "Frequency-based Search-control in Dyna"
_ICLR.cc/2020/Conference — Accept (Poster)_

### Official Review · AnonReviewer3 · 2019-10-21
**Official Blind Review #3**

**Rating:** 6

**Review:**

Summary:
This paper basically built upon [1]. The authors propose to do sampling in the high-frequency domain to increase the sample efficiency.  They first argue that the high-frequency part of the function is hard to approximate (i.e., needs more sample points) in section 3.1. They argue that the gradient and Hessian can be used to identify the high-frequency region. And then they propose to use g(x)=||gradient||+||Hessian || as the sampling metric as illustrated in Algorithm 1. To be noticed that, they actually hybrid the proposed metric (6) and the value-based metric (7, proposed in [1]) in their algorithm.

Strength:
Compared to [1], their experiment environment seems more complicated (MazeGridWorld vs. GridWorld).
Figure 3 shows that their method converges faster than Dyna-Value.
Figure 5 is very interesting. It shows that their method concentrates more on the important region of the function.

Weakness:
In footnote 3: I don't see why such an extension is natural.
In theorem 1, why the radius of the local region has to be?
Theorem1 only justifies the average (expectation) of gradient norm is related to the frequency. The proposed metric $g$, however, is evaluated on a single sample point. So I think if adding some perturbations to $s$ (and then take the average) when evaluating $g$ will be helpful.
The authors only evaluate their algorithm in one environment, MazeGridWorld.
I would like to see the experiment results of using only (6) as the sampling rule.
What kind of norm are you using? (||gradient||, ||hessian||)
Why $g$ is the combination of gradient norm and hessian norm? What will be the performance of using only gradient or hessian?
Figure 4(b), DQN -> Dyna

Reference:
[1] Hill Climbing on Value Estimates for Search-control in Dyna

**Experience Assessment:**

I have read many papers in this area.

**Review Assessment: Checking Correctness Of Derivations And Theory:**

I assessed the sensibility of the derivations and theory.

**Review Assessment: Checking Correctness Of Experiments:**

I assessed the sensibility of the experiments.

**Review Assessment: Thoroughness In Paper Reading:**

I read the paper at least twice and used my best judgement in assessing the paper.

---

> ### Author Response · Authors · 2019-11-11
> **Address all main issues**
>
> We appreciate your insightful and constructive feedback. We hope our response addresses the main weaknesses as you pointed out. We uploaded the most recent version of our paper.
>
> Why the theorem is natural to extend to second order derivative?
> In the updated paper, we formally wrote down the theoretical result of second order connection for clarification. In previous version, we say it is natural because that the reasoning line of deriving second order connection is exactly the same as we do for the first-order derivation. For your convenience, here we summarize the key steps: 1) use a local Fourier transform to express a function locally (i.e. within an unit ball); 2) calculate the gradient/Hessian norm based on this local Fourier transform; 3) take integration over the unit ball of the gradient/Hessian norm to build the connection with the local frequency distribution $\pi_{\hat{f}}$ and function energy. In Appendix A.2, after the definition of f_x(y), we take gradient one more time to acquire one row of the Hessian matrix. Then we still use the complex conjugate for the square of Frobenius norm of the Hessian matrix, which can be calculated by summing over the square of l_2 of all the rows in the Hessian. We use l2 norm for vector and Frobenius norm for matrix.
>
> The radius does not have to be one, but assuming it to be one does not lose generality. Since we cannot talk about the frequency of a single point, the key idea is to characterize the frequency of a function locally and we use a ball (hypersphere) to define the local domain.
>
> The suggestion of using some sort of sample average is very interesting. In fact, the idea of adding perturbation to s is implemented in both the previous work by Pan et al. and in our current work, please see Appendix A.6 eq(14)(15). With this noise added to each gradient step, then we can connect the hill climbing to Langevin dynamics, then the points along the gradient ascent trajectory can be roughly thought of as Gibbs distribution where the density is proportional to g(s), and hence it is justifiable that our search-control queue provides states with higher density in the high frequency region. For more details, we refer readers to the section 4.2 in the paper Hill Climbing on Value Estimates for Search-control in Dyna by Pan et al.
>
> Notice that sampling using only rule (8)(b) (rule (6) in the previous version) is the algorithm from the work by Pan et al, which is indeed included in our paper and is marked as Dyna-Value. If you actually mean only using rule (8)(a) (rule (7) in the previous version), we explain the choice in 2nd paragraph in section 4. Theorem 1 tells us that the frequency is connected with both the function magnitude and the gradient norm. We want to avoid finding those states with very small values which are unlikely to visit under an optimal policy. As a result, we choose initial states which have high value for gradient ascent, then the gradient ascent should help further move to higher frequency region.

---

### Official Review · AnonReviewer4 · 2019-11-05
**Official Blind Review #4**

**Rating:** 6

**Review:**

This paper proposes a new way to select states from which do do transitions in dyna algorithm (which trains policy from model experience as if it was a real experience). It proposes to look for states where frequency of value function as a function of a real valued state is large, because these are the states where the function is harder to approximate. The paper also shows that such frequency is large where the gradient of the function is large in magnitude which allows for finding such states in practice. In more detail, similar to previous algorithms, this algorithm keeps both an experience replay buffer as well as another buffer of states (search-control queue) and uses a hill climbing strategy to find states with both higher frequency and higher value. The paper tests the algorithm on toy domains - the mountain car and a maze with doors.

The idea of using the magnitude of the gradient as an exploration signal is not new - “go to places where agent learns more”. In this paper, such signal is not used as a reward but for finding states from which to do training updates. It is also nice that the paper provides a good relation (with explanation) between this signal and the frequency of the value function. The paper is clearly written. One drawback is that the main computations are only tractable in toy domains - it would be good if they discussed how to use this with general neural model with large state spaces (e.g. states obtained with an RNN).

Detailed comments:
- In the abstract it says “…searching high frequency region of value function”. At this point it is not clear what function we are considering - what is on the axes? - a time, state, value? (value on y axis and a real valued state on the x as it turns out later).
- Same at line end-7 on page 2
- End of section 2: and states with high value (as you do later).
- A demonstration experiment: Why do you fit linear regression to a sin function? Why not at least one layer NN?
- Page 5 top: You reason that Hessian learns faster - why not just squaring gradient norm?
- Section 4: Hessian is intractable for general neural network unless you have a toy domain - does it work with just the gradient? This is an important point if this is to be scaled. May be you can also discuss how to compute the gradient of the norm of the gradient.


**Experience Assessment:**

I have published one or two papers in this area.

**Review Assessment: Checking Correctness Of Derivations And Theory:**

N/A

**Review Assessment: Checking Correctness Of Experiments:**

I assessed the sensibility of the experiments.

**Review Assessment: Thoroughness In Paper Reading:**

N/A

---

> ### Author Response · Authors · 2019-11-11
> **Address all main issues**
>
> We appreciate your insightful and constructive feedback. We uploaded the most recent version of our paper.  We hope we address your concern well.
>
> The sinus experiment: Mentioning “linear” was a slip. We indeed use a neural network, as explained in detail in the 2rd paragraph, (Appendix) A.6 (A.5 in the previous version).
>
> Computational cost. We discuss two solutions here. The first one is exactly as you mentioned. We show some results of only using first-order information in the updated paper Appendix A.5. We write g(s) as the sum of the gradient norm and hessian norm for generality and for the purpose of better matching our theoretical result. The gradient w.r.t. state of gradient norm can be calculated as: $\frac{\partial ||\nabla_s V(s)||^2}{\partial s} = \frac{\partial^2 V(s)}{\partial s^2} \frac{\partial V(s)}{\partial s} = H_V(s) \nabla_s V(s)$. Note that the Hessian vector product can be calculated by backpropogating twice, known as an efficient algorithm. We recommend "Fast Exact Multiplication by the Hessian", Pearlmutter’93 for reference.
>
> The second solution should be more interesting. Note that our hill climbing strategy can be used w.r.t feature. That is, we can do gradient ascent on latent feature instead of the observation variables. Enforcing the action value to be some simpler function in the feature can greatly reduce the computational cost. This is particularly useful for the purpose of either handling large observation space (i.e. an image) or handling partial observability. There are several existed works studying feature-to-feature models. Once we acquire feature vectors through search-control, those models can be used for planning update. We believe this is a promising future direction.
>
> About squaring gradient norm. In general, Hessian norm and squaring gradient norm are not equivalent. Figure 2 in section 3.2 should throw some insight about this. The mathematical derivation of building the connection between Hessian norm and local frequency resembles that between gradient norm and local frequency. We added the second order theoretical connection into the updated paper.
>
> We would like to briefly discuss existed works of exploring uncertain regions. To our best knowledge, in those works, the uncertainties are characterized w.r.t. the learning parameters (not the state variables of the value function). Concretely, given V_\theta(x), the previous works concern about uncertainty of parameters \theta, while we concern about where the training $x$s should come from based on the nature of true V (this is independent of \theta, though we have to use \theta in implementation as the true V is unknown). Please let us know if we miss any reference, we would be happy to add those.

---

### Official Review · AnonReviewer5 · 2019-11-05
**Official Blind Review #5**

**Rating:** 6

**Review:**

Disclaimer:
This is an emergency review that I got assigned on Nov 5th, so I unfortunately had only a few hours to assess the paper.

Summary:
This paper proposes a new mechanism for “search control”, that is, choosing the planning-start-states from which to roll out the model, in the context of Dyna-style updates. The key idea is to favour updates to states where the the value functions changes a lot, because those need more information to become accurate.
The topic is very interesting, and not studied enough, the proposed method looks novel, and the experimental results look promising. However, the clarity of the paper is not great, the theoretical result (Theorem 1) seems to be incorrect, the narrative from Claude Shannon to Joseph Fourier to gradient norms to practical choices is somewhat confusing, the empirical results are not really conclusive, and despite a large appendix there are missing implementation details. I think this paper is currently one or two major revisions away from the acceptance threshold.

Comments:
1. Theorem 1, equation (3): consider a constant function f(x) = K. Then its derivative is zero, so the left-hand side is zero, yet the right-hand side is the product of two strictly positive factors (each of them proportional to K^2), so how can the equation hold?
2. Sine experiment: How can you reasonably do *linear* regression onto a sine function? What are your input features?
3. Figure 2 is odd: panels (b) and (c) look very similar in how they bias the sampling toward the left side, yet the performance difference (a) is very stark, how come? Also, how can there be such a high contrast in (b)/(c) if 60% of all samples are chosen uniformly, as stated in the text?
4. The paper has numerous grammatical mistakes, to the extent it becomes difficult to read. I know this can happen in deadline mode, but please revise the draft thoroughly (special call-out to the dozens of missing definite/indefinite articles and plural forms). Also, use “\citep” where appropriate.
5. The objective g as the sum of a gradient norm and a Hessian norm seems odd, as these terms have completely different scales, so usually one of them will dominate, can you explain and motivate this further, and compare empirically to the two terms in isolation?
7. For the prioritizedER baseline, what variant and hyper-parameters are you using?
8. Please describe “out of boundary” mentioned in Algorithm 1.
9. Section 5.2 states that the “variance of the evaluation curve” is smaller, indicating robustness, yet Figure 3(a) appears to have high instability in the (early) learning curve?
10. Section 5.2 states that Figure 5 should show that the bottleneck areas are sampled more densely, but that’s a dubious claim. Please quantify the claim or drop it.
11. Figure 4(b) has oddly wide error-bars for DQN-Value, which looks suspiciously like a single failed run. Can you add a plot to the appendix with median/quantiles instead of mean/std statistics?


--------------
Update Nov 17: most of my concerns have been addressed, and I have thus increased my rating.





**Experience Assessment:**

I have published one or two papers in this area.

**Review Assessment: Checking Correctness Of Derivations And Theory:**

I assessed the sensibility of the derivations and theory.

**Review Assessment: Checking Correctness Of Experiments:**

I assessed the sensibility of the experiments.

**Review Assessment: Thoroughness In Paper Reading:**

I read the paper at least twice and used my best judgement in assessing the paper.

---

> ### Author Response · Authors · 2019-11-10
> **Address all main issues**
>
> We thank you for reviewing our paper within such a short notice and providing us with very specific and valuable feedback. Our most recent version of paper is uploaded. We believe our paper is well-written and quite clear. We hope that if your concerns are addressed, you update your score accordingly.
>
> Theorem 1: The theorem is correct. Eq. (3) (in the new version, Eq. (5)) holds in your example.
> The Fourier transform of a constant function is Dirac’s delta function at frequency zero.
> Because of the property of the Dirac’s delta function that
> $\int f(x) \delta(x - x0) dx = f(x0)$, we see that the second integral in the RHS is proportional to $\int \delta(k - 0) ||k||^2 dk = ||0|| = 0$. Therefore, both sides are zero.
>
> The sinus experiment: Mentioning “linear” was a slip. We indeed use a neural network, as explained in detail in the 2rd paragraph, (Appendix) A.6 (A.5 in the previous version).
>
> Figure 2. We updated that figure and report the concrete proportions of points fall in high frequency region. Please see Empirical Demonstration, Sec 3.2. In fact, (c) has higher density around the spikes and we quantified this in the updated paper. Around the spikes (which have large second-order derivative magnitude), the function changes sharply and it is more difficult to generalize and hence we need more samples around those parts. The experiment is easy to reproduce according to details in the 2nd paragraph, A.6. We also want to mention that the main purpose of that figure is not to compare (b) and (c); instead, we simply want to show that the distribution of using either first order and second order prioritization can lead to high density on the frequency region.
>
> Regarding the objective g. We added them in A.5. But we want to clarify that it does not matter which one would dominate or whether using one of them would be better, since we characterize local frequency through both first-order and second-order derivatives, so the g(s) formulation is general and matches with our theoretical result (notice that in the updated paper, we formally established the second-order connection). An even more general form to write g(s) is some weighted sum of the first-order and second-order terms (then your suggestion is a special case). However, we do not need such complications in our experiments. Your phrase “one of them would dominate” is, in fact, a benefit. As you mentioned, the two terms can have vastly different scales. Regions which have low (or even zero) gradient magnitude may have high Hessian magnitude, and vice versa. Hence, using the combination can help the hill climbing process in case that one of the term vanished at some point.
>
> PrioritizedER baseline. Sorry for missing the detail in the paper. We use the proportional variant with sum tree data structure. We use prioritized ER without importance ratio bias correction but half of mini-batch samples are uniformly sampled from ER as a strategy for bias correction. This matches with our own algorithm. We add these details into the paper in A.6. In fact, our algorithm can easily outperform ER and PrioritizedER. As we increase the number of planning steps, most samples from the ER buffer would be updated sufficiently, hence PrioritizedER or ER should show limited improvement. In contrast, our algorithm can utilize the generalization power of the value function to acquire unvisited states during search-control, hence it can benefit from the larger number of planning updates. Figure 5 from https://arxiv.org/pdf/1906.07791.pdf may be a good visualization for the difference between ER and search-control queue.
>
> Out of boundary: This refers to being outside the valid subset of the states. In many problems, we know that states are within a proper subset of R^d, e.g., [0,1]^d. If the hill climbing procedure generates a state outside that subset, we restart the hill climbing process. This is the same as discussed by Pan et al., “Hill Climbing on Value Estimates for Search-control in Dyna,” IJCAI, 2019.
>
> About “variance of the evaluation curve” in section 5.2. Please notice that our claim about “the variance of the evaluation curve” refers to Figure 4(b) and not Figure 3(a). In the former, the shaded areas are noticeably different. In Figure 3(a), we do not see how our algorithm appears to have high instability in early learning curve, and we are unsure how you would define “early learning” and “high instability”. In fact, in term of variance, the behaviour of our algorithm in both Figure 4(b) and Figure 3(a) should be considered as consistent: it has a lower standard error than other competitors.
>
> Section 5.2, points in bottleneck areas. We add concrete counts in the figure caption.
>
> Figure 4(b). The wide error-bars for Dyna-Value is exactly what we expected. Without sufficient samples from the bottleneck areas, the agent is likely fail to pass the holes. Some of the runs for Dyna-Value failed (not a single run).

---

### Decision · Program_Chairs · 2019-12-19

**Decision:**

Accept (Poster)

**Comment:**

The reviewers are unanimous in their evaluation of this paper, and I concur.